# Water-Yield Relationship and Vegetative Growth of Wonderful Young Pomegranate Trees under Deficit Irrigation Conditions in Southeastern Italy

**Annalisa Tarantino *** 📧, **Laura Frabboni and Grazia Disciglio**

Department of Agriculture Food, Natural Science and Engineering, University of Foggia, 70100 Foggia, Italy; laura.frabboni@unifg.it (L.F.); grazia.disciglio@unifg.it (G.D.)
***** Correspondence: annalisa.tarantino@unifg.it

**Abstract:** This investigation was carried out through three successive seasons (2017, 2018, and 2019) on young pomegranate trees of the Wonderful cultivar to study the effect of four different irrigation treatments (100%, 75%, 50%, and 25% of crop evapotranspiration—*ET*c) on vegetative growth and the water–yield relationship. The study was conducted in Foggia (Apulia region, Southern Italy), an agricultural area characterized by strong wind speeds and scarce water resources. The results showed the effects of the different irrigation levels and seasons on the vegetative growth and fruit yield characteristics. The cumulative trunk diameter, the annual shoot growth, the number of fruits per tree, and the yield decreased from the full water restitution (100% *ET*c) to the severe water restriction (25% *ET*c). The weight and the size of fruits decreased significantly with the restriction of water volumes applied to the crop. A linear relationship between water consumption and yield ($R = 1.00$ in 2018 and 1.21 in 2019; $n = 12$) was found. The water use efficiency (WUE) gave no statistical differences among irrigation treatments. The yield response factors (*Ky*: 1.06 in 2018 and 0.99 in 2019) showed the sensitivity of pomegranates to water deficits.

**Keywords:** pomegranate; irrigation regime; shoot growth; trunk growth, yield response factor; water use efficiency

## 1. Introduction

Modern water-saving agriculture aims to develop sustainable agriculture by focusing on improving water use efficiency and on the use of appropriate irrigation variables. Different plant species have different drought tolerances, the identification and evaluation of which is subject to environmental conditions. Pomegranate *(Punica granatum* L., *Punicaceae)* is fairly drought resistant, and cultivation is mainly confined to semi-arid, mild-temperate areas including most of the Mediterranean countries, which are characterized by water scarcity [1,2]. It is a domesticated tree with relatively short juvenility [3] whose fruits are of high nutritional and economic value [4]. In Italy, the pomegranate is cultivated in such southern semiarid regions as Sicily, Apulia, Basilicata, and Calabria, where water is a scarce resource. Particularly in the Apulia region, over the last decade, the area of pomegranate cultivation has increased significantly, passing from 8 ha in 2009 [5] to about 1000 ha (60% of the total national cultivation area) in cultivation area today [6], in which Wonderful is the cultivar most planted. To cope with water scarcity for this crop, there is an increasing move to more efficient technological innovation and irrigation management approaches such as the drip irrigation method and soil plastic mulching applied around pomegranate plants, which produces a positive effect on fruit yield and quality [7–9]. Moreover, many authors have already studied the pomegranate response under deficit irrigation (DI) strategies, with the aim of improving its development under water stress conditions, minimizing production losses, and keeping or even improving fruit quality. Some studies have demonstrated that restricting irrigation produced a decrease in photosynthesis and

stomatal conductance, affecting the plants vegetative growth [10] and the growth of new buds, shoots, and trunks [11,12], while the highest irrigation level stimulated vegetative growth by increasing the shoot length, the number of leaves per shoot, and the leaf area [13]. Another study [14] showed that dendrometric responses varied with species and plant age. Moreover, Intrigliolo et al. [15] showed that DI with 50% of the crop water used during the entire season resulted in yields similar to full irrigation, whereas other authors [16,17] indicated that DI reduces the total yield, the number of fruits per tree, and the size of the fruit. Finally, other reports concluded that the use of a DI strategy throughout the growing season or at specific phenological stages is useful [18,19]. The discrepancy among these studies can be attributed to the different pomegranate varieties, the agro-climatic conditions, and environmental factors [20]. Currently, no definitive solid criteria exist for deciding the optimum timing of irrigation, probably because of the diversity of factors involved, such as evaporative demand, soil characteristics, the soil water status at any precise moment, the crop phonological stage, etc. This underscores the necessity of conducting irrigation research under local conditions [11]. However, the improvement in pomegranate production by using DI requires knowing the important agronomic factors in the planned irrigation under water-limiting conditions such as the field water use efficiency (WUE), that is, the ratio of the yield of the crop to the total amount of water used in the field [21] and the yield response to the water factor ($Ky$), which represents the effect of a reduction in evapotranspiration on yield losses [22]. To the best of our knowledge, no information exists in this study's area, which is also characterized by strong wind speeds, regarding pomegranate water management aspects.

Taking into account all of the above-mentioned considerations, the main objective of this study was to evaluate the effects of deficit irrigation on the water use efficiency (WUE) and the yield response factor as well as on vegetative growth and the yield of young pomegranate trees. Finally, it should be noted that, from this same experiment, the effects of different irrigation regimes on the physicochemical and phytochemical characteristics of the fruits have been reported in a recently published paper [23].

## 2. Materials and Methods

### 2.1. Experimental and Climatic Conditions

The field trial was carried out at Foggia (41°27′08′′ N, 15°31′56′′ E and an altitude of 54 m above see level. during three consecutive seasons (2017, 2018, and 2019) with the Wonderful pomegranate cultivar, which corresponded to plants 2, 3, and 4 years old. The commercial orchard was grown from cutting and trained to a vase shape and was drip-fertirrigated and mulched with a plastic sheet along the tree rows. The drip irrigation system comprised a single pipe with drippers at with the 21 h$^{-1}$ flow rate spaced every 40 cm. During the growing seasons, standard agronomic practices were applied. The soil texture was clay-loam [24] with an effective depth over 120 cm. The site of the research was in a typical semi-arid zone, characterized by a Mediterranean climate, which is classified as an accentuated thermomediterranean climate [25], with temperatures that may fall below 0 °C in the winter and exceed 40 °C in the summer. Rainfall is unevenly distributed throughout the year and is mostly concentrated in the winter months with a long-term annual average of 559 mm [26].

During each experimental season, from April to October, daily climate data were recorded at the nearest meteorological station, which was a few kilometers from the experimental area. The data were used to calculate the reference evapotranspiration ($ET_O$) by the Penman–Monteith equation [27]. The shaded area (*SA*) of the trees and the crop coefficient (*Kc*) were also estimated and used to calculate the maximum crop evapotranspiration ($ET_C$), the actual crop evapotranspiration ($ET$a), and other irrigation parameters.

### 2.2. Irrigation Treatments

Four different irrigation treatments (full water restitution of 100% of the $ET$c and deficit irrigation of 75%, 50%, and 25% of the $ET$c), starting from the bud break stage to fruit maturity, were performed. Each watering was applied by keeping a constant interval (7 days) and by changing the water volume from 100% to 75%, 50%, and 25% of the $ET$c net of rains [28,29] The irrigation volumes applied were measured using in-line water meters installed on each experimental plot. The experimental design was a randomized complete block design with three replicates per treatment. Each plot had three rows with 8 trees in each. The three central trees in the middle row were used for data collection. All experimental trees were surrounded by guard trees in order to minimize the effect of irrigation.

### 2.3. Determination of the ETc

The $ET$c was calculated weekly by a climatological approach, multiplying $ET_O$ by $Kc$ [27]. Considering that the orchard had not reached the final size, the $Kc$ values were corrected by the shaded area ($SA$) fraction of the soil at solar noon as the actual surface subjected to irrigation, and they were calculated by Equation (1):

$$SA_a = \pi R^2 N/100 \tag{1}$$

where, $SA_a$ = the actual shaded area; $R$ = the average radius of the canopy's circumference, as measured on selected plants; $N$ = the number of trees per hectare. The corrected $Kc_a$ was calculated by Equation (2) [30,31]:

$$Kc_a = Kc \times (1 - (SA_a/SA_{max})^{0.5} \tag{2}$$

where, $Kc_a$ = the actual crop coefficient; $Kc$ = the FAO crop coefficient; $SA_a$ = the actual percentage of the shaded area; $SA_{max}$ = the potential percentage of the soil shaded area (80%) of the mature pomegranate orchard ($N$ = the planning density equal to 606 trees per ha).

Ultimately, the actual evapotranspiration ($ET$a) was calculated by Equation (3):

$$ETc = ETo \times Kc_a \tag{3}$$

Weekly values of $ET_O$, $Kc_a$, and $SA_a$ were used to obtain the weekly values of $ET$c, starting from the bud break stage (April) to the fruit maturity stage (September). A water balance was carried out to determine the irrigation required.

### 2.4. Determination of the Yield Water Use Efficiency (YWUE) and the Yield Response Factor (Ky)

The yield water use efficiency (YWUE) was calculated in each treatment from the ratio of the total yield (in kilograms) to the amount of irrigation water applied (including rainfall) to the crop ($Kg \cdot m^{-3}$ water). The yield response factor ($Ky$), which links the relative yield decrease to the relative evapotranspiration deficit, is expressed by the following Formulation (4) [22]:

$$Ky = (1 - Ya/Ym)/(1 - ETa/ETm) \tag{4}$$

where, $Ya$ ($Kg\ ha^{-1}$) and $Ym$ ($Kg\ ha^{-1}$) are the actual and maximum yields (obtained from the fully irrigated treatment), respectively; $ETa$ ($m^3 \cdot ha^{-1}$) and $ETm$ ($m^3 \cdot ha^{-1}$) are the actual and maximum water consumptions (of the fully irrigated treatment), respectively.

### 2.5. Vegetative Growth Parameters

The trunk diameter (TD) was determined in three seasons, whereas the shoot length (SL) was only determined in the second and third seasons. Both were measured on three plants in each irrigation plot. The TD was measured two times in each year, at bud break (in April) and at the end of growth season (in October). It was measured at a marked point, 50 cm above the ground level, using the Vernier digital caliper, and expressed in

millimeters. The annual trunk growth (ATG) and the trunk cumulative growth (TCG) were also calculated. The SL was determined at the end of each growing season on four tagged one-year-old shoots in the four direction points (East, West, North, and South) using a tape and expressed in centimeters.

### 2.6. Yield, Morpho-Pomological Characteristics of the Fruits

In the first experimental season (2017), all the flowers were thinned by hand, to guarantee the adequate development of trees, whereas, in the second and third years, fruiting was allowed, and the harvests were carried out on 18 October, 2018 and on 29 October, 2019, respectively. At harvest time, the fruits per tree for each plot were counted and weighted. The average number of fruits per tree and the yield (ton·ha$^{-1}$) were calculated. Samples of 12 fruits from each irrigation treatment of each replicate were randomly collected. Each fruit was weighed (g) and measured in length (mm) and diameter (mm).

### 2.7. Statistical Analysis

The results were evaluated with one-way ANOVA using JMP® software version 8 (SAS Institute Inc., Cary, NC, USA) and average values were compared with the Tukey test. Standard deviations (SD) were calculated using Excel software of the Office 2007® suite (Microsoft Corporation, Redmond, WA, USA). Percentage values were transformed to arcsine prior to analysis of variance.

## 3. Results and Discussion

### 3.1. Weather Conditions and Water Consumption

In Table 1, the climatic data and the *ET*c in the three experimental seasons (April–September) are reported. The weather conditions between seasons were quite similar, whose values of the total *ET*o (a representative climate evaporative parameter) were 1080.7, 1061.1, and 1044.7 mm in 2017, 2018, and 2019, respectively. During each growing season, the *ET*c increased with increasing canopy and evaporative air demand, reaching the peak in July. The total seasonal *ET*c also increased over the years, with increasing plant development from 310.4 to 375.4 to 518.9 mm in 2017, 2018, and 2019, respectively. In each season, the cumulated water requirement values (*ET*c-0.70 *p*) became positive in June. Therefore, the differentiation of irrigation treatments in each year began from the flower-petals-opening fall stage (fruit setting) to the fruit-ripening stage.

**Table 1.** The monthly mean maximum and minimum temperatures ($T_{max}$ and $T_{min}$), the relative air humidity ($RH_{max}$ and $RH_{min}$), the wind speed ($W_s$), the radiation (*Rad*), the total precipitation (*p*), the reference evapotranspiration ($ET_O$), the percentage of the shaded area of the trees ($SA_a$), the crop coefficient ($K_C$), the crop evapotranspiration (*ET*c), and the cumulated water requirement (*ET*c-0.70 *p*) during the 2017, 2018, and 2019 seasons.

| Month | $T_{max}$ | $T_{min}$ | $RH_{max}$ | $RH_{min}$ | $W_s$ | *Rad* | *p* | *ET*o | $SA_a$ | $Kc_a$ | *ET*c | (*ET*c-0.70 *p*) |
|---|---|---|---|---|---|---|---|---|---|---|---|---|
| | (°C) | (°C) | (%) | (%) | (m·s$^{-1}$) | (W·m$^{-2}$) | (mm) | (mm) | % | (mm) | (mm) | (mm) |
| | | | | | | | 2017 | | | | | |
| April | 21.3 | 9.7 | 90.1 | 47.0 | 3.1 | 172.8 | 82.0 | 138.9 | 5.5 | 0.20 | 27.8 | −29.6 |
| May | 26.1 | 12.1 | 95.1 | 44.1 | 2.4 | 232.4 | 94.1 | 170.3 | 8.6 | 0.30 | 51.1 | −14.8 |
| June | 33.2 | 17.6 | 83.8 | 30.4 | 2.7 | 262.5 | 1.0 | 205.0 | 10.6 | 0.30 | 61.5 | +60.5 |
| July | 34.1 | 20.4 | 85.5 | 31.0 | 3.0 | 329.3 | 24.0 | 215.6 | 11.2 | 0.30 | 64.7 | +40.7 |
| Aug | 35.0 | 19.2 | 78.0 | 26.8 | 2.7 | 314.6 | 14.6 | 201.9 | 11.2 | 0.30 | 60.6 | +50.4 |
| Sept | 26.1 | 14.9 | 80.0 | 34.1 | 3.5 | 189.7 | 72.0 | 149.0 | 11.2 | 0.30 | 44.7 | −5.7 |
| Mean | 29.3 | 15.7 | 85.4 | 35.6 | 2.9 | 250.2 | | | 9.7 | 0.28 | | |
| Total | | | | | | | 287.7 | 1080.7 | | | 310.4 | 100.9 |

**Table 1.** *Cont.*

| Month | $T_{max}$ | $T_{min}$ | $RH_{max}$ | $RH_{min}$ | $W_s$ | *Rad* | *p* | *ET*o | $SA_a$ | $Kc_a$ | *ET*c | (*ET*c-0.70 *p*) |
|---|---|---|---|---|---|---|---|---|---|---|---|---|
| | (°C) | (°C) | (%) | (%) | (m·s$^{-1}$) | (W·m$^{-2}$) | (mm) | (mm) | % | (mm) | (mm) | (mm) |
| | | | | | | 2018 | | | | | | |
| April | 21.3 | 12.9 | 94.6 | 37.6 | 2.8 | 235.3 | 54.0 | 159.8 | 8.7 | 0.20 | 31.9 | −6.2 |
| May | 26.1 | 13.4 | 95.2 | 49.1 | 2.4 | 275.8 | 58.3 | 170.6 | 15.4 | 0.30 | 51.2 | +10.4 |
| June | 30.0 | 12.1 | 89.5 | 40.3 | 3.4 | 289.6 | 88.2 | 180.7 | 21.3 | 0.40 | 72.3 | +10.3 |
| July | 33.3 | 19.6 | 83.6 | 35.4 | 3.0 | 318.7 | 16.8 | 206.3 | 22.6 | 0.40 | 82.5 | +70.8 |
| Aug | 32.7 | 20.1 | 71.3 | 28.3 | 2.1 | 285.7 | 39.1 | 187.3 | 22.6 | 0.40 | 74.9 | +47.5 |
| Sept | 29.1 | 17.1 | 81.3 | 30.0 | 3.7 | 193.6 | 80.0 | 156.4 | 22.6 | 0.40 | 62.6 | +6.6 |
| Mean | 28.7 | 15.9 | 85.9 | 36.8 | 2.9 | 266.5 | | | 18.9 | 0.35 | | |
| Total | | | | | | | 366.4 | 1061.1 | | | 375.4 | 140.3 |
| | | | | | | 2019 | | | | | | |
| April | 20.6 | 8.2 | 94.4 | 51.0 | 3.7 | 190.2 | 40.3 | 131.6 | 18.0 | 0.30 | 39.5 | +11.3 |
| May | 21.3 | 10.2 | 95.3 | 56.3 | 4.0 | 232.9 | 86.7 | 150.5 | 25.4 | 0.40 | 60.2 | −0.49 |
| June | 33.2 | 17.5 | 85.9 | 35.1 | 3.7 | 252.2 | 9.2 | 200.3 | 25.8 | 0.50 | 110.0 | +103.6 |
| July | 33.7 | 19.5 | 84.0 | 33.9 | 3.7 | 258.8 | 30.0 | 207.0 | 30.2 | 0.55 | 113.8 | +92.8 |
| Aug | 34.8 | 20.3 | 79.9 | 33.9 | 3.6 | 225.6 | 5.7 | 198.9 | 30.5 | 0.55 | 109.4 | +105.4 |
| Sept | 29.5 | 16.8 | 88.7 | 42.6 | 3.6 | 175.5 | 3.8 | 156.4 | 30.5 | 0.55 | 86.0 | +83.3 |
| Mean | 28.9 | 15.4 | 88.0 | 42.5 | 3.7 | 222.5 | | | 26.7 | 0.47 | | |
| Total | | | | | | | 175.7 | 1044.7 | | | 518.9 | 395.9 |

### 3.2. Applied Irrigation

Table 2 shows the irrigation volumes, whose values obviously increased over the years as the canopy of the plants and the relative *Kc* and *ET*c increased.

**Table 2.** Water applied (mm) and watering (*n*.) with different levels of irrigation.

| Year | Seasonal Irrigation Volume (mm) | | | | Watering |
|---|---|---|---|---|---|
| | $ETc_{100}$ | $ETc_{75}$ | $ETc_{50}$ | $ETc_{25}$ | (*n*.) |
| 2017 | 101 | 83 | 55 | 27 | 11 |
| 2018 | 140 | 105 | 70 | 35 | 12 |
| 2019 | 395 | 296 | 197 | 124 | 14 |

The seasonal irrigation volumes for the 100%, 75%, 50%, and 25% *ET*c treatments were: 101, 83, 55, and 27 mm, respectively, with 11 watering events in 2017; 140, 105, 70, and 35 mm, respectively, with 12 watering events in 2018; and 395, 296, 197 and 99 mm, respectively, with 14 watering events in 2019 (Table 2).

### 3.3. The Effects of Irrigation Treatments on Vegetative Growth

The annual trunk growth (Figure 1) was the lowest in the first year, whose diameter values were not statistically different among the irrigation treatments (average = 7.0 mm). In the second and third years, $ETc_{100}$ and $ETc_{75}$ treatments showed a significantly higher growth than in the first year. $ETc_{50}$ and $ETc_{25}$, although showing some differences among years, were not statistically different.

Trunk cumulative growth (TCG) (Figure 2) gradually increased with some differences among the irrigation treatments and the two seasons. During the first season, the increase was almost uniform in all the treatments. It seems that all irrigation treatments supplied enough water for plants because of their small size and low water requirements. However, during the second season, and especially during the third season, the pomegranate trees had a different growth, showing a lower capacity for tolerating water restrictions [32,33]. After three consecutive seasons, the largest TCG (42.6 mm) was noticed in the $ETc_{100}$ treatment, although it was not significantly different from the $ETc_{75}$ (39.0 mm) and $ETc_{50}$ (38.8 mm) treatments but was statistically different from the $ETc_{25}$ (35.5 mm) treatment. These results, which show the sensitivity of stem growth of pomegranate trees to water

restriction, were similar to those obtained by Bugueño et al. [34]. Additionally, previous studies [11,12,35] conducted on young almond, lemon, olive, and cherry trees showed that water restrictions reduced the cumulative trunk growth rate.

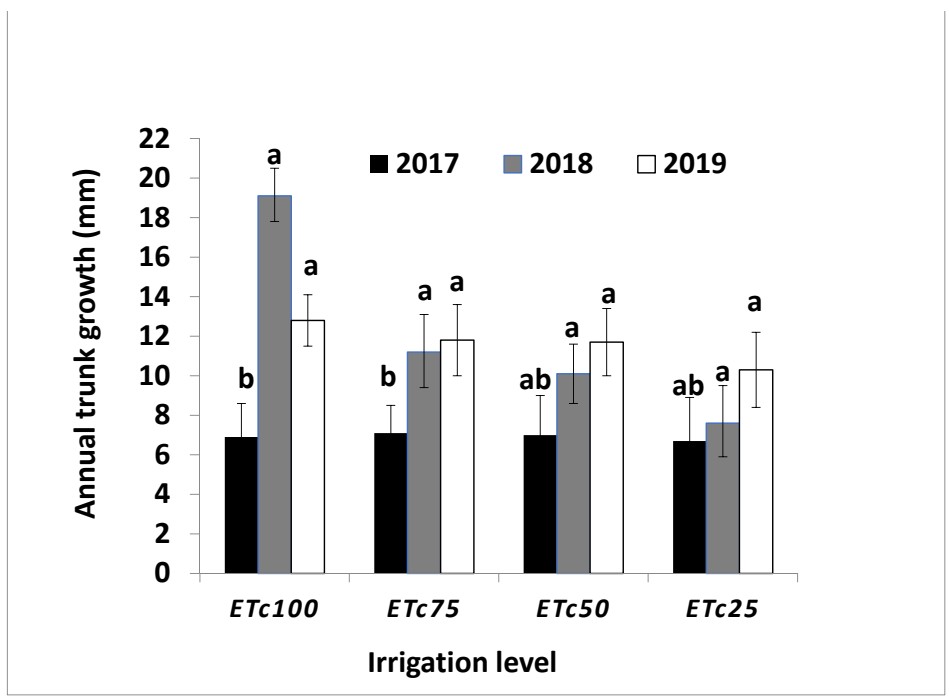

**Figure 1.** Annual trunk growth of the irrigation treatments in the 2017, 2018, and 2019 growing seasons. Different letters indicate no significant differences among seasons at $p \leq 0.05$, according to the Tukey test.

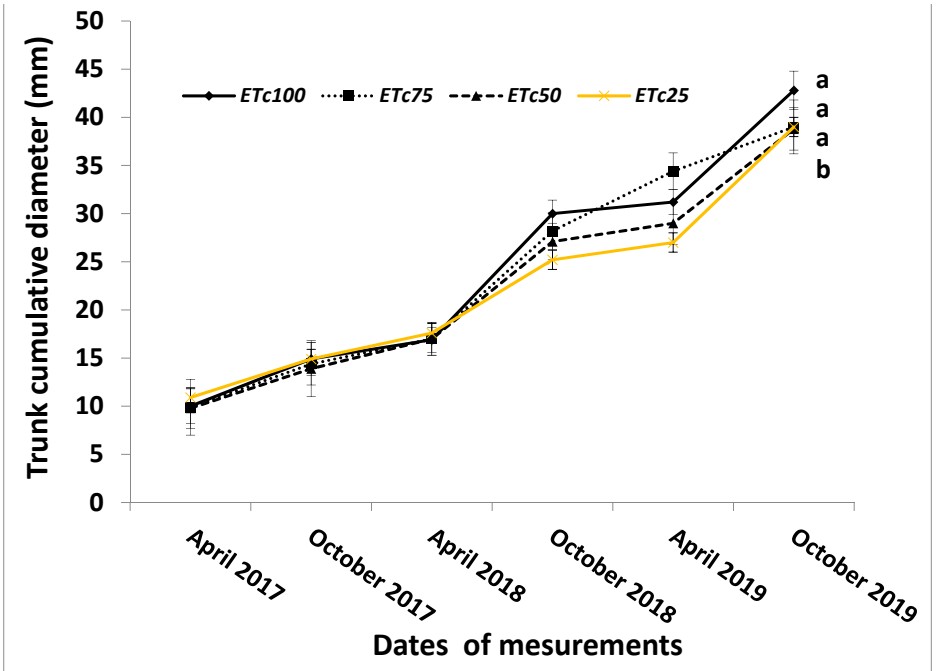

**Figure 2.** Trunk cumulative growth for each irrigation treatment during the three seasons from April 2017 to October 2018. Irrigation treatments with different letters at the end of the three seasons were significantly different at $p \leq 0.05$, according to the Tukey test.

As for the shoot development, monitored only in the second and third years, no significant differences were noted among the four spatial directions of the trees. A decrease in length as irrigation water application decreased was noted, although there were no significant differences between treatments because of the high variability of the data (Table 3). These results are in harmony with those obtained by Khattab et al. [13]. Significant higher values were obtained in 2019 (on average 42.1 cm) than in 2018 (on average 12.9 cm) (Figure 3).

**Table 3.** Annual shoot growth (cm) as influenced by the four direction points and different levels of irrigation in the 2018 and 2019 seasons.

| Treatment | Year | North | East | South | West | Mean |
|---|---|---|---|---|---|---|
| $ETc_{100}$ | 2018 | 22.6 ± 9.1 b | 14.3 ± 3.7 b | 13.3 ± 2.9 b | 9.0 ± 3.6 b | 14.8 ± 6.9 b |
| | 2019 | 52.3 ± 24.8 a | 44.3 ± 6.0 a | 42.6 ± 2.5 a | 55.7 ± 9.3 a | 48.7 ± 10.6 a |
| $ETc_{75}$ | 2018 | 18.3 ± 7.5 b | 9.1 ± 6.9 b | 15.3 ± 5.5 b | 14.6 ± 9.8 b | 14.3 ± 7.4 b |
| | 2019 | 51.6 ± 11.5 a | 34.7 ± 13.8 a | 48.3 ± 7.6 a | 52.0 ± 3.6 a | 46.7 ± 9.1 a |
| $ETc_{50}$ | 2018 | 17.0 ± 8.2 b | 11.0 ± 5.7 b | 9.7 ± 2.1 b | 11.3 ± 1.5 b | 12.2 ± 4.3 b |
| | 2019 | 43.3 ± 9.2 a | 24.3 ± 9.2 a | 43.3 ± 10.4 a | 45.6 ± 10.2 a | 39.1 ± 7.2 a |
| $ETc_{25}$ | 2018 | 6.0 ± 0.3 b | 12.0 ± 0.6 b | 12.7 ± 0.6 b | 10.7 ± 4.0 b | 10.3 ± 1.3 b |
| | 2019 | 41.3 ± 3.5 a | 30.6 ± 10.6 a | 35.0 ± 5.0 a | 28.3 ± 10.6 | 33.8 ± 8.7 a |

Deviations followed by a different letter within the lines and columns were significantly different at $p \leq 0.05$, according to the Tukey test.

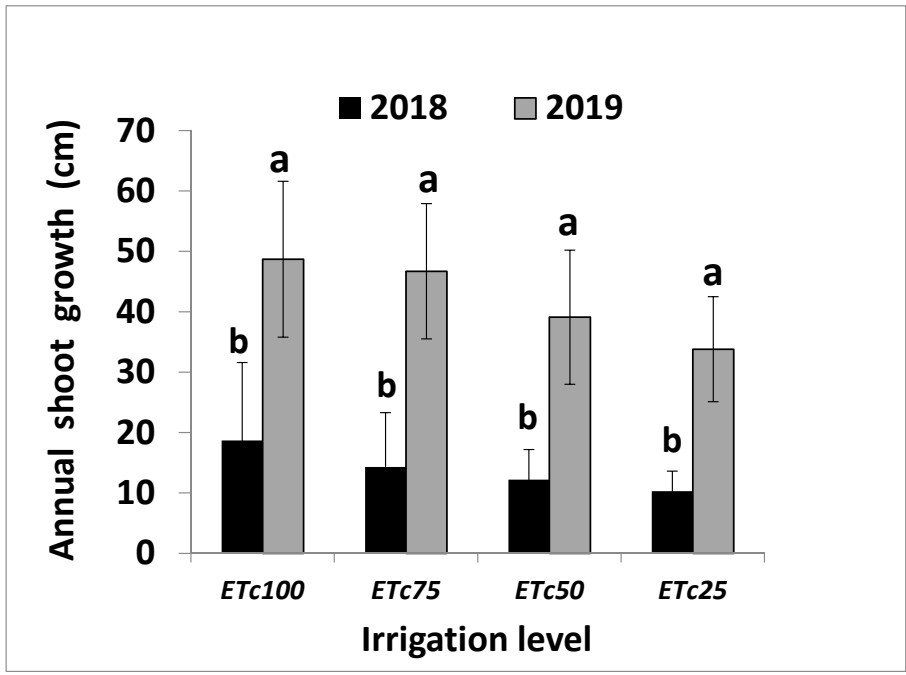

**Figure 3.** The annual shoot growth of the irrigation treatments in the 2018 and 2019 growing seasons. Different letters indicate significant differences between seasons at $p \leq 0.05$, according to the Tukey test.

### 3.4. The Effect of Irrigation Treatments on Yield

The average number of fruits per tree ($n_o$) and the yields (t·ha$^{-1}$) in all the irrigation treatments were significantly higher in 2019 (averages of 18.9 $n_o$ and 3.6 t·ha$^{-1}$, respectively) than in 2018 (averages of 9.3 $n_o$ and 2.5 t·ha$^{-1}$, respectively) (Table 4). These differences, obviously, depended on the age of the plants. As regards the yield, it should be noted that its increase in 2019 compared to 2018 was less than proportional to the number of fruits per plant. This was due to the lower average weight of the fruit recorded in this last year, as reported in Table 5. As a general trend, the aforementioned parameters decreased

with reducing irrigation levels, albeit with some differences between the years. In 2018, the $ETc_{100}$ treatment gave the highest number of fruits per tree (11.6), which was not significantly different from the $ETc_{75}$ (10.5) and the $ETc_{50}$ (8.9) treatments, but which was statistically different from the $ETc_{25}$ (6.5) treatment. Instead, in 2019, the $ET_{100}$ treatment provided the highest number (26.3), which was significantly different from the $ETc_{75}$ (19.7) treatment, which, in turn, was statistically different from the $ETc_{50}$ (15.7) and the $ETc_{25}$ (14.0) treatments. Similar to the number of fruit, in 2018, the highest yield of the $ETc_{100}$ treatment (3.67 t·ha$^{-1}$) was not significantly different from that of the $ETc_{75}$ (2.79) treatment but was statistically different from the $ETc_{50}$ (2.24) and the $ETc_{25}$ (1.31) treatments. In 2019, the $ETc_{100}$ treatment's highest values (5.94 t·ha$^{-1}$) were significantly different from the $ETc_{75}$ (4.10) treatment, which, in turn, was statistically different from the $ETc_{50}$ (2.97) and the $ETc_{25}$ (1.57) treatments. These results showed that, in the first productive year (2018), the pomegranate was not very sensitive to moderate (75% and 50%) water stress treatments but was very sensitive to severe stress (25%), which caused a significant yield reduction. In the successive seasons, any stress caused a significant yield reduction. These results are in harmony with those obtained in other studies [16,36], but are in contrast with those of Centofanti et al. [21], who reported that the DI strategies did not significantly affect the yield in the Wonderful cultivar.

**Table 4.** Water consumption and the effect of irrigation levels on the number of fruits per tree, the yield, and the water use efficiency of the Wonderful pomegranate cultivar in the 2018 and 2019 seasons.

| Treatment | Seasonal Water Consumption (Including Rainfall) (m$^3$·ha$^{-1}$) | | Fruits Per Tree | | Fruit Yield | | WUE | |
|---|---|---|---|---|---|---|---|---|
| | 2018 | 2019 | 2018 | 2019 | 2018 | 2019 | 2018 | 2019 |
| $ETc_{100}$ | 1766 | 4216 | 11.6 ± 1.8 a | 26.3 ± 2.6 a | 3.67 ± 0.40 a | 5.94 ± 0.77 a | 2.08 ± 0.33 | 1.41 ± 0.8 |
| $ETc_{75}$ | 1416 | 3130 | 10.5 ± 1.3 a | 19.7 ± 1.2 b | 2.79 ± 0.51 a | 4.10 ± 0.15 b | 1.97 ± 0.30 | 1.31 ± 0.10 |
| $ETc_{50}$ | 1066 | 2146 | 8.8 ± 1.2 ab | 15.7 ± 1.7 c | 2.22 ± 0.40 b | 2.97 ± 0.38 c | 2.08 ± 0.25 | 1.37 ± 0.18 |
| $ETc_{25}$ | 716 | 1166 | 6.5 ± 1.3 b | 14.0 ± 1.4 c | 1.32 ± 0.35 c | 1.57 ±0.19 d | 1.85 ± 0.28 | 1.35 ± 0.25 |

Means ± std. deviations followed by a different letter within the columns were significantly different at $p \leq 0.05$, according to the Tukey test.

**Table 5.** The effect of different irrigation treatments on the pomegranate fruits' morphological characteristics in the 2018 and 2019 seasons.

| Treatment | Fruit Weight (g) | | Fruit Diameter (mm) | | Fruit Length (mm) | |
|---|---|---|---|---|---|---|
| | 2018 | 2019 | 2018 | 2019 | 2018 | 2019 |
| $ETc_{100}$ | 522.7 ± 90.7 a | 496.2 ± 107.9 a | 95.6 ± 4.7 a | 94.2 ± 5.0 a | 83.8 ± 4.7 a | 85.0 ± 7.0 a |
| $ETc_{75}$ | 432.7 ± 95.0 ab | 358.4 ± 43.5 a | 89.1 ± 8.2 ab | 88.2 ± 3.6 ab | 80.2 ± 6.5 ab | 76.0 ± 3.7 ab |
| $ETc_{50}$ | 378.7 ± 110.3 ab | 361.7 ± 102.1 a | 84.7 ± 9.0 ab | 85.9 ± 7.0 ab | 76.3 ± 9.6 ab | 75.4 ± 5.8 ab |
| $ETc_{25}$ | 332.5 ± 85.6 b | 304.1 ± 107.5 a | 81.8 ± 6.7 b | 80.4 ± 8.1 b | 70.3 ± 7.3 b | 73.4 ± 4.2 b |

Means ± std. deviations followed by a different letter within the lines were significantly different at $p \leq 0.05$, according to the Tukey test.

Furthermore, as it is seen in Figure 4, a linear crop-water-consumption–yield relationship was found in both years ($R$ = 0.85 in 2018 and 0.99 in 2019; $n$ = 12), whose yield reduction was directly related to the reduction in water consumption. This results differ from those obtained from Aydin et al. [37] who found a second-degree polynomial relationship between the irrigation water and the yield.

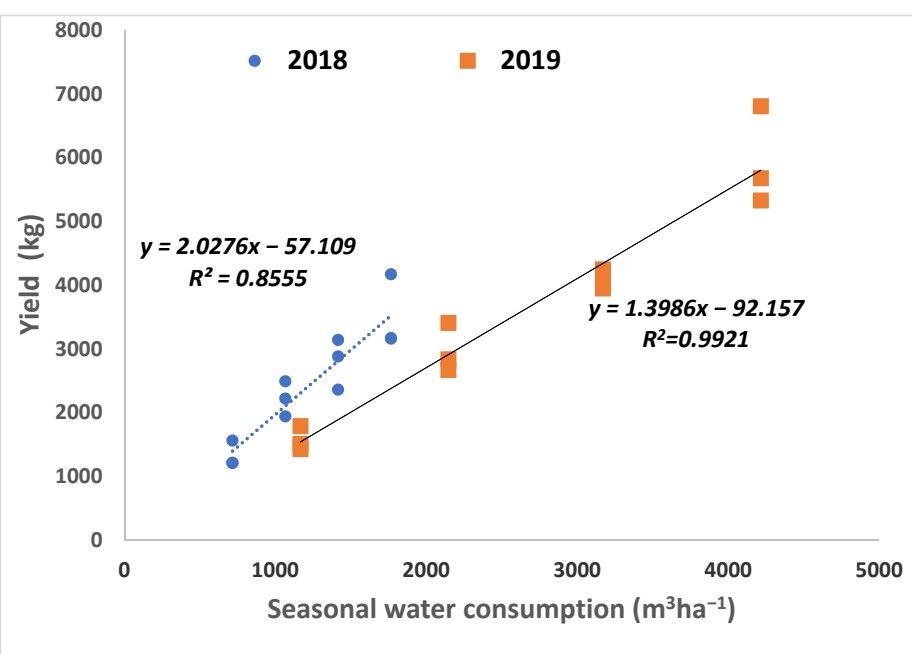

**Figure 4.** Linear water production functions for pomegranates subjected to water deficits occurring during the 2018 and 2019 growing seasons (*n* = 12).

### 3.5. The Effect of Irrigation Treatments on Morphological Characteristics of Fruit

The average weight, diameter, and length of fruits showed significantly lower values in 2019 compared to 2018. In both years, these parameters showed the highest values in 2019 compared to 2018. In both years these parameters showed the hhighest values in $ETc_{100}$ treatment, although these were not significantly different from the $ETc_{75}$ and the $ETc_{50}$ treatmetns and were not different from the $ETc_{25}$ ones (Table 5).These results are in accordance with a previous study [38] that reported a reduction in the fruit size in pronounced water deficit levels. These results obtained on the physical prosutctive response to irrigation can provide information relevant to production and industrial purposes. For the latter, in addition to the qualitative aspects of the juice (content of bioactive compounds, organic acids, sugars, etc.), reported in a prvious study [23], the knowledge of the mor-phololological properties of fruit is useful in designing optimal equipment for harvesting, trasporting, grading, cleaning, storing and packaging as well as processing the whole fruit into other derived products.

### 3.6. The Water use Efficiency (WUE) and the Yield Response Factor (Ky)

As a general trend, Table 4 indicates that the WUE values were slightly higher in 2018 (they varied between 1.85 and 2.08 kg/m$^3$) than those in 2019 (they varied between 1.31 and 1.41 kg/m$^3$). However, in both years, no statistical differences in the WUE among irrigation treatments were noted, although, in both years, slightly higher values were obtained in the fully irrigated treatment. These results differ from other previously obtained results [13,39], which indicated significantly higher WUE values in trees under highly irrigated levels.

As for the yield response factor, the results showed a slightly higher value in 2019 (1.21) than in 2018 (1.00) (Figure 5). In any case, according to the *Ky* obtained in this study, it can be said that young Wonderful pomegranate plants are sensitive to drought stress during the growth period. Our *Ky* values were higher than that (0.81) reported by Taha [40], which expressed a moderate tolerance to water stress, but were much lower than that (1.57) reported by Aydin et al. [37] for the Zivzik variety, which expressed a high sensitivity to water stress. These differences may be due to factors other than water, such as nutrients, different environments, different cultivars, and the age of the trees, which also affects the yield response to water [22].

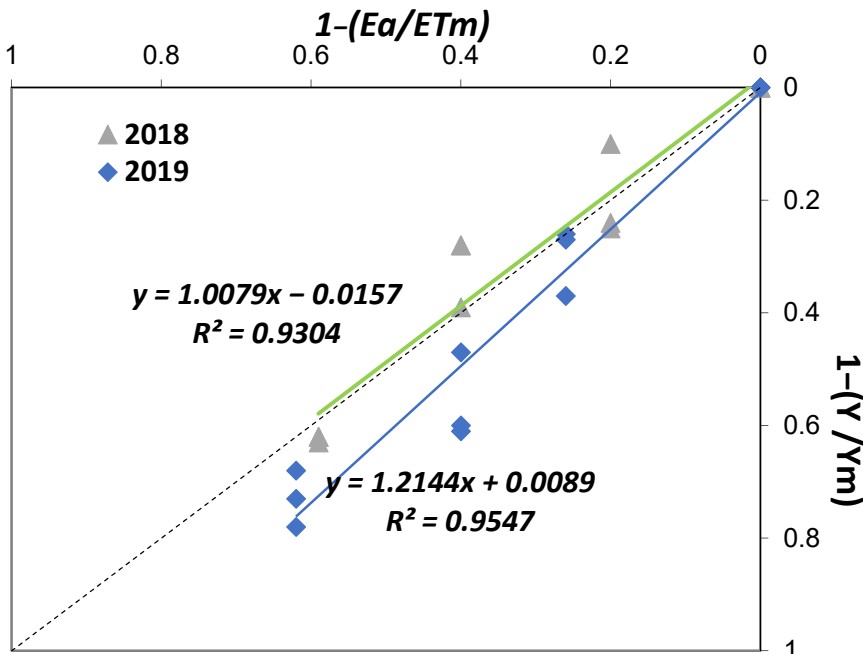

**Figure 5.** Relationships of the yield decrease ($1 - Y_a/Y_m$) vs. the $ET$ decrease ($1 - ET_a/ET_m$). The yield response factor ($K_y$): 1.00 in 2018 and 1.21 in 2019 ($n$ = 12).

## 4. Conclusions

This study investigated the adoption of a DI strategy (restitution of 100%, 75%, 50%, and 25% of the $ET_C$) during three successive growing seasons on pomegranates to evaluate the effects on vegetative growth, yield, the water use efficiency, and the yield response factor. The results showed that the plants at the end of the third experimental trial reached the significantly highest cumulative trunk diameter by the restitution of 100% of the $ET_c$, while it gradually decreased with the increase in water stress treatments. Additionally, the shoot length, on average, decreased as irrigation water applications decreased, although insignificant differences were noted among treatments because of the high variability of the data. As a general trend, the average number of fruits per tree, the fruit size, and the yield decreased with reducing irrigation levels, among which the severe water restriction (25% $ET_c$) always provided the statistically lowest values. A linear crop-water-consumption–yield relationship was found, where yield reduction was directly related to reduced water use. Moreover, although slightly higher WUE values were obtained in the fully irrigated plants, no statistical differences among irrigation treatments were noted. The results of the yield response factor to the reduction in water use showed values of 1.00 and 1.21 in the 2018 and 2019 seasons, respectively. This means that, under our experimental conditions, the young Wonderful pomegranate plants were sensitive to drought stress during the growth period.

In conclusion, based on the above-mentioned results, young Wonderful pomegranate trees in semiarid areas, also characterized by strong speed winds, require full water restitution throughout the dry season to reach optimal growth and yield. However, considering that the effect of DI on pomegranate productive responses has been only recently studied and that results from this and several other previously cited studies are contradictory, longer-term studies are needed in semiarid areas to better predict physiological responses to DI strategies relative to the productivity of pomegranate.

**Author Contributions:** Conceptualization, A.T. and G.D.; data curation, A.T., L.F. and G.D.; formal analysis, A.T., G.D.; investigation, A.T. and L.F.; methodology, A.T., L.F. and G.D.; supervision, A.T.; validation A.T. and G.D.; visualization, A.T.; writing—original draft, A.T; writing—review

and editing, A.T., L.F. and G.D. All authors have read and agreed to the published version of the manuscript.

**Funding:** This research did not receive any specific grant from funding agencies in the public, commercial, or not-for-profit sectors.

**Acknowledgments:** We sincerely thank Dario Croella for his kind hospitality at his farm and his effective cooperation in carrying out project activities.

**Conflicts of Interest:** The authors declare that they have no conflicts of interest.

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
