# Peer review of "Water-Yield Relationship and Vegetative Growth of Wonderful Young Pomegranate Trees under Deficit Irrigation Conditions in Southeastern Italy"

_horticulturae, doi:10.3390/horticulturae7040079_

Round 1

Reviewer 1 Report

This topic is crucial under unpredictable climate condition es[ecially when water is scarse.

I have no complain on almost all in this manuscript.

However, I cannot understand why irrigaation volume gets increasing in Table 2.

The restricted irrigation in 2019 has higher volume than that of 2017.

Furthermore, fruit yield did not get doubled as seasonal water comsumption and fruits per trees did in Table 4.

And, WUE decreased.

There should be CLEAR explaining and interpretation on these.

Last but not least, WUE should be spelled out or inserted after water use efficiency in legend. 

Author Response

I sent the cover letter for the reviewer 1

Reviewer 2 Report

The manuscript focuses on describing the effect of four irrigation regimes on pomegranate growth and productivity. The experiment is well designed and the results clear. However, in my opinion, a better explanation of the results meaning, and applications is missing. A reduction in water availability take to a reduction in productivity, those results could be expected, but which are the implication of these results to the industry?

More in detail:

Introduction

Line 68-72: this sentence is too long, please split the information in 2 or 3 sentences.

Material and methods

Line 96: Please, specify which are ‘other irrigation parameters’. If those ‘other irrigation parameters’ are not described or not relevant for this study, please eliminate that from the sentence.

Which software was used for the graphs? It seems the graphs have different styles, which makes the comparison between treatments more difficult.

Results and Discussion

Line 220-221: The equation of linear relation need to be added in the text or in caption of the figure.

Fig 4 probably do not need a color line connecting each point. Only the points with the linear relation.

Author Response

I sent the cover letter for the Reviewer 2

Round 2

Reviewer 1 Report

Authors improved their manuscript significantly.

Thus, I suggest that it should be considered to be accepted.